# Sustainable intensification for a larger global rice bowl

Shen Yuan[1], Bruce A. Linquist[2], Lloyd T. Wilson[3], Kenneth G. Cassman[4], Alexander M. Stuart[5], Valerien Pede[5], Berta Miro[5], Kazuki Saito[6], Nurwulan Agustiani[7], Vina Eka Aristya[8], Leonardus Y. Krisnadi[9], Alencar Junior Zanon[10], Alexandre Bryan Heinemann[11], Gonzalo Carracelas[12], Nataraja Subash[13], Pothula S. Brahmanand[14], Tao Li[15], Shaobing Peng[1]✉ & Patricio Grassini[4]✉

Future rice systems must produce more grain while minimizing the negative environmental impacts. A key question is how to orient agricultural research & development (R&D) programs at national to global scales to maximize the return on investment. Here we assess yield gap and resource-use efficiency (including water, pesticides, nitrogen, labor, energy, and associated global warming potential) across 32 rice cropping systems covering half of global rice harvested area. We show that achieving high yields and high resource-use efficiencies are not conflicting goals. Most cropping systems have room for increasing yield, resource-use efficiency, or both. In aggregate, current total rice production could be increased by 32%, and excess nitrogen almost eliminated, by focusing on a relatively small number of cropping systems with either large yield gaps or poor resource-use efficiencies. This study provides essential strategic insight on yield gap and resource-use efficiency for prioritizing national and global agricultural R&D investments to ensure adequate rice supply while minimizing negative environmental impact in coming decades.

[1] National Key Laboratory of Crop Genetic Improvement, Hubei Hongshan Laboratory, MARA Key Laboratory of Crop Ecophysiology and Farming System in the Middle Reaches of the Yangtze River, College of Plant Science and Technology, Huazhong Agricultural University, Wuhan 430070 Hubei, China. [2] Department of Plant Sciences, University of California-Davis, One Shields Ave., Davis, CA 95616, USA. [3] Texas A&M AgriLife Research Center, Beaumont, TX 77713, USA. [4] Department of Agronomy and Horticulture, University of Nebraska-Lincoln, Lincoln, NE 68588, USA. [5] International Rice Research Institute, DAPO Box 7777 Metro Manila, Philippines. [6] Africa Rice Center (AfricaRice), 01 B.P. 2551, Bouake 01, Côte d'Ivoire. [7] Indonesian Center for Rice Research, Sukamandi 41256, Indonesia. [8] Assessment Institute of Agricultural Technology (AIAT) Central Java, Ungaran 50552, Indonesia. [9] Assessment Institute of Agricultural Technology (AIAT) East Java, Malang 65152, Indonesia. [10] Universidade Federal de Santa Maria, Avenida Roraima n° 1000, 97105-900 Santa Maria, Rio Grande do Sul, Brazil. [11] EMBRAPA Arroz e Feijão, Zona Rural GO-462, Santo Antônio de Goiás, Goias 75375-000, Brazil. [12] Rice Program, National Institute of Agricultural Research (INIA)-Road 5, km 386, Tacuarembó, Uruguay. [13] ICAR-Indian Institute of Farming Systems Research, Modipuram 250110 Uttar Pradesh, India. [14] ICAR-Indian Institute of Water Management, Bhubaneswar 751023 Odisha, India. [15] Applied GeoSolutions, DNDC Applications Research and Training, Durham, NH 03824, USA. ✉email: pgrassini2@unl.edu; speng@mail.hzau.edu.cn

Rice (*Oryza sativa* L.) is the main staple food for nearly half the world's population, accounting for 21% of global calorie intake while using 11% of global cropland[1,2]. Rice yields have increased steadily since the onset of the Green Revolution in the late 1960s, driven by adoption of high-yielding rice cultivars, intensive use of agricultural inputs, and investments in irrigation infrastructure, extension education services, and subsidies[3,4]. Global rice consumption is projected to increase from 480 million tons (Mt) milled rice in 2014 to nearly 550 Mt by 2030, driven by both population increase and economic growth in developing countries[5]. However, there are a number of concerns about the future sustainability of rice cropping systems. First, yield growth rates have slowed down, and even reached a plateau, in some major rice-producing regions such as California (USA), China, Indonesia, and South Korea[6,7]. Second, negative environmental impact is a concern because rice production consumes 30%, 14%, and 10%, respectively, of global use of irrigation water, fertilizers, and pesticides[8–11], leading in some cases to negative environmental impacts. Furthermore, rice cultivation is an important source of anthropogenic greenhouse gas (GHG) emissions, accounting for 30 and 11% of global agricultural methane ($CH_4$) and nitrous oxide ($N_2O$) emissions, respectively[12,13]. Third, high labor requirements and other associated costs make rice production less attractive to farmers in some regions[9], especially where national governments are becoming reluctant to provide price support mechanisms and subsidies[14].

Increasing concerns about loss of natural habitats and conservation of biodiversity emphasize the importance of producing more rice on existing cropland and to do so while improving the efficiencies of energy, nutrient, water, and other inputs in a process called sustainable intensification[15–17]. Prioritizing investments on agricultural research and development (R&D) at national to global scales to increase rice production and minimize the environmental impact requires information on current yield gaps and resource-use efficiencies. The yield gap is defined as the difference between yield potential and average farmer yield. Yield potential is determined by solar radiation, temperature, water supply, cultivar traits, and, in the case of water-limited crops, also by precipitation and soil properties and landscape characteristics influencing water balance[18,19]. Achieving ca. 70–80% of yield potential is a reasonable target for farmers who have good access to markets, inputs, and extension services[3,20]. Further closure of the yield gap is difficult due to decreased return on additional inputs and labor, and the high degree of sophistication in crop management required to accommodate the spatial and temporal variation in weather, soil properties, pest pressure, etc[20]. In contrast, regions with large yield gaps have largest potential to increase current yield through use of existing cost-effective agronomic practices. In relation to the environmental impact, metrics on resource use (energy, water, nutrients, pesticides, etc.) on an area basis do not account for differences in yield level among cropping systems, which in turn affects land requirements to produce adequate national or global rice supply. Instead, metrics relating resource use with yield (e.g., yield-scaled energy use, nutrient balances, etc.) are more appropriate for global assessments because they allow proper benchmarking of input use for a given yield level[21–23]. All else equal, the magnitude of the environmental impact is closely related to the efficiencies in use of energy, water, nutrients, and pesticides[23,24]. To summarize, information on yield gaps and resource-use efficiencies can help identify regions with greatest potential to increase production, reduce environmental impact, or both, and guide agricultural R&D prioritization.

There have been efforts to benchmark rice yield gaps and/or resource-use efficiencies for individual countries or regions[24–27]. In contrast, we are not aware of any global assessment of yield gaps and resource-use efficiencies for rice cropping systems that can serve to prioritize agricultural R&D investments to increase rice production while reducing associated environmental impact. Herein, we present the results from a global assessment of rice yield gaps and resource-use efficiencies based on yield potential reported in the Global Yield Gap Atlas (www.yieldgap.org) and actual yield and agricultural input data collected across 32 rice cropping systems in 18 rice-producing countries, accounting for 51% of global rice harvested area (Supplementary Tables 1–3). Pathways to narrow down existing yield gaps and reduce the negative environmental impacts are discussed.

## Results

**Current yield gaps in rice cropping systems.** Across cropping systems, the number of rice crops grown on the same piece of land during a 12-month period can range from one in non-tropical regions to three in tropical environments (Supplementary Figs. 1 and 2). Here we report metrics on a per-crop basis by averaging values across the rice crops within each cropping system where more than one rice crop is grown each year. Metrics computed on an annual basis are provided in the Supplementary Information. Similarly, average values reported in this study are weighted according to the annual rice harvested area in each cropping system. At a global scale, yield potential averaged 9.5 Mg ha$^{-1}$ crop$^{-1}$, ranging from 5.9 to 14.8 Mg ha$^{-1}$ crop$^{-1}$ across the 32 rice cropping systems included in our analysis (Supplementary Fig. 3). Average yield potential is higher in non-tropical regions than in tropical regions (9.9 versus 8.8 Mg ha$^{-1}$ crop$^{-1}$). Lower productivity per crop of tropical rice is more than compensated by higher cropping intensity as tropical regions have longer growing seasons that allow up to three rice crops each year in the same field (Supplementary Figs. 1 and 2). As a result, rice systems in tropical areas have greater annual potential productivity than in non-tropical regions (15.3 versus 12.2 Mg ha$^{-1}$ year$^{-1}$) (Supplementary Fig. 3A). In our study, all rainfed cropping systems are in lowland environments, except for rainfed upland rice in Brazil. Despite growing in flooded soil during much of the growing season, rainfed lowland rice can be exposed to water deficit and/or excess flooding during a portion of the cropping season[28], as it is the case of rainfed lowland rice systems in South-East Asia, leading to lower and less stable yield potential compared with irrigated rice (Supplementary Figs. 3 and 4).

Expressing average actual farmer yields as percentage of yield potential helps normalize farmer yields across cropping systems with different climate background and water regimes, providing an objective measure of the degree to which rice farmers efficiently utilize solar energy and water resources. At a global scale, average rice yield represents 57% of yield potential, with a wide range of yield gaps across rice systems (Fig. 1 and Supplementary Fig. 5). For example, irrigated rice systems in Egypt, northern China, Australia, and California have reached ca. 75% of the yield potential. At the other end of the spectrum, average yields are low for rainfed lowland rice in Sub-Saharan Africa and rainfed upland rice in northern Brazil and represent 20–40% of the yield potential. About two-thirds of the total rice harvested area included in our 32 cropping systems have yields <75% of yield potential; the latter is considered a reasonable yield gap closure target for farmers[20]. Overall, our analysis indicates substantial room to increase global rice production on existing planted area via improved agronomic management.

**Benchmarking resource-use efficiencies.** We look at key environmental and resource-use metrics associated with rice cropping systems, including global warming potential (GWP), water supply (sum of irrigation plus in-season precipitation), pesticide use,

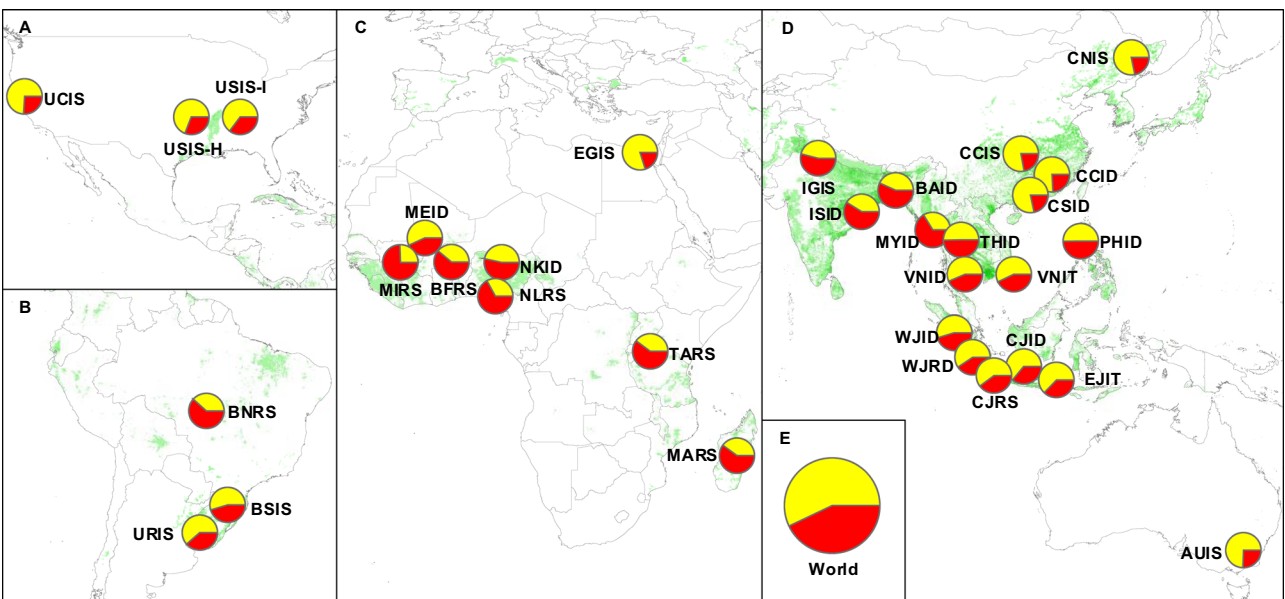

**Fig. 1 Map showing the rice systems assessed in this study and their associated yield gap (red portions of pie charts) and actual yield (yellow portions of pie charts) expressed as percentage of the yield potential.** Panels correspond to **A** North America, **B** South America, **C** Africa, **D** Asia and Australia, and **E** world. Rice area distribution is shown in green (SPAM maps[77]). Cropping system code consists of region (first two letters), water regime (third letter), and rice cropping intensity (fourth letter). Regions: Australia (AU); Bangladesh (BA); northern and southern Brazil (BN and BS, respectively); Burkina Faso (BF); central, northern, and southern China (CC, CN, and CS, respectively); Egypt (EG); Indo-Gangetic Plains and southern India (IG and IS, respectively); central, east, and west Java, Indonesia (CJ, EJ, and WJ, respectively); Madagascar (MA); Segou and Sikasso, Mali (ME and MI); Myanmar (MY); Kano and Lafia, Nigeria (NK and NL); Philippines (PH); Tanzania (TA); central Thailand (TH); southern USA and California (US and UC); Uruguay (UR); Vietnam (VN). In the case of the southern USA, hybrid (H) and inbred rice (I) are also distinguished. Water regime: irrigated (I) and rainfed (R). Cropping intensity: single (S), double (D), and triple (T). Description of each rice cropping system and associated yield potential and yield gap are provided in Supplementary Figs. 1–5 and Supplementary Tables 1–9. Data are provided in Source Data.

nitrogen (N) balance, and labor inputs. Despite a strong positive correlation between the degree of yield gap closure and total input use per unit area expressed as GWP, high-yield systems have lower GWP on a yield-scaled basis due to higher resource-use efficiency (Fig. 2A, B). For example, high-yield systems in Egypt, northern China, Australia, and California have larger GWP per hectare, but smaller yield-scaled GWP than low-yield, low-input systems in Sub-Saharan Africa. An implication from this finding is that, to reach a given grain production target, low-input systems have larger land requirement compared to high-input, high-yield systems which, in turn, can lead to a larger negative environmental impact due to conversion of fragile natural ecosystems such as wetlands and forests for rice production. These results are consistent regardless of whether GWP is considered on a per-crop or annual basis (Supplementary Fig. 6).

There is no relationship between the degree of yield gap closure and water supply ($p = 0.50$), probably because water supply is sufficient to meet crop water requirement in most cropping systems (Fig. 2C and Supplementary Fig. 6; Supplementary Table 4). For a similar degree of yield gap closure, there is a large range in water supply due to differences in climate among cropping systems[28]. For example, water supply is ca. 1.2x larger in the semiarid climate of California, USA compared with the humid southern USA. Similarly, there is large variation in yield gap at any given water supply, with rainfed rice exhibiting a larger gap compared with irrigated rice. The yield-scaled water supply follows a relationship with the degree of yield gap closure similar to that for yield-scaled GWP ($r = -0.60$; $p < 0.01$), with smallest values corresponding to cropping systems with small yield gaps (Fig. 2D). Many of these systems are located in semiarid environments (e.g., California, Egypt, and Australia), where rice production takes places in fields purposely selected based upon

soil type in order to minimize percolation losses, and where crops are likely to be fully irrigated, with little precipitation to supplement crop water demand, and with high yield potential due to high solar radiation (Fig. 2D and Supplementary Fig. 4). Given the low production risk and favorable conditions, these systems are also likely to have a smaller yield gap. Assessing the long-term sustainability of irrigated rice systems in water-scarce environments would benefit from expanding the analysis to larger spatial scales (e.g., watershed) and accounting for recharge rates and stream flows. Likewise, rice grown during the wet season in some of these cropping systems may have water in excess of storage capacity, making a low yield-scaled water supply irrelevant for these systems. While incomplete, our study makes a first step on this direction by benchmarking the efficiency in using water resources to produce rice at field scale.

In the case of pesticides, there is a positive relationship between yield (as percentage of potential) and the number of applications ($r = 0.51$; $p < 0.01$) (Fig. 2E). It is difficult to interpret this relationship considering likely differences in edaphic and climatic environments and the severity of biotic stresses. Higher pesticide use in cropping systems with small yield gaps is possibly related to greater pest and disease pressures as a result of large and denser leaf canopies that are achieved with improved plant nutrition[29]. Likewise, systems with high cropping intensity (i.e., double and triple rice) in tropical areas receive a larger number of insecticide and fungicide applications per crop (up to nine) as in the case of Indonesia and Vietnam (Fig. 2E). There are also labor cost considerations. In contrast to tropical systems, where rural labor wage is low and weeds are mainly removed manually, chemical control prevails in non-tropical cropping systems (Supplementary Table 5). Due to these interrelationships, the relationship between yield-scaled pesticide applications and yield

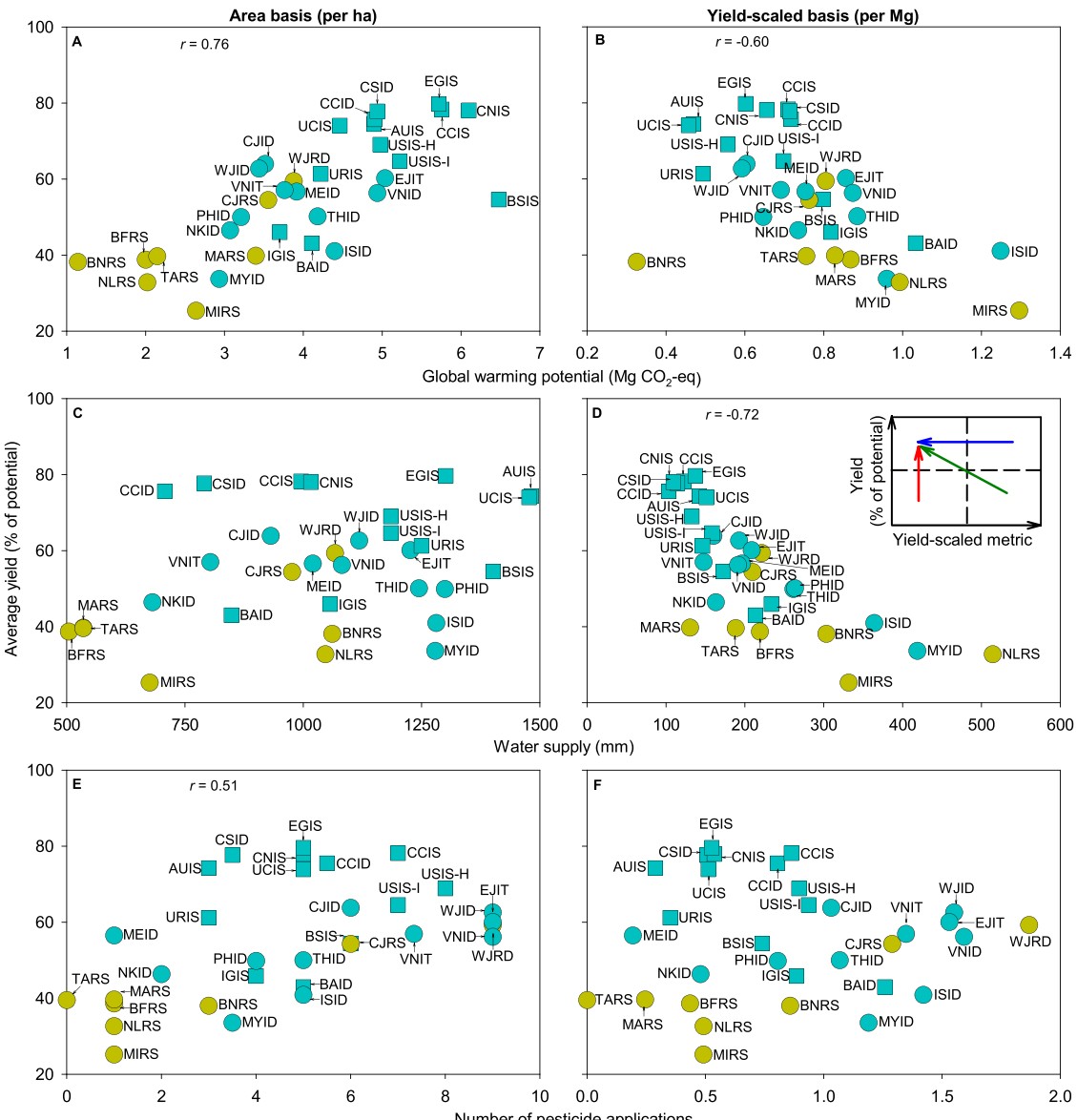

**Fig. 2 Yield, global warming potential, water supply, and number of pesticide applications on both area and yield-scaled basis across 32 rice cropping systems.** Panels show average rice yield (expressed as percentage of yield potential) versus **A**, **B** global warming potential, **C**, **D** water supply (irrigation plus in-season precipitation), and **E**, **F** number of pesticide applications. Global warming potential, water supply, and number of pesticide applications are shown on an area (**A**, **C**, **E**) or yield-scaled basis (**B**, **D**, **F**). Each point represents the average for the rice crops in each cropping system (typically two or three for irrigated rice in tropical regions and one or two for rainfed rice in tropical regions and for irrigated rice in non-tropical regions). Symbol type and color are used to distinguish tropical versus non-tropical regions (circles and squares, respectively) and irrigated versus rainfed systems (blue and yellow, respectively). Inset in **D** shows hypothetical pathways to improve yield and/or reduce environmental impact. Pearson's correlation coefficient ($r$) is shown only when the association between variables is significant (two-tailed Student's $t$-test; $p < 0.01$; $n = 32$ cropping systems). Cropping system codes are shown in the caption to Fig. 1. Data are provided in Source Data.

gap closure is not as clear as for GWP and water supply (Fig. 2F), although a similar trend is apparent when cropping systems from Sub-Saharan Africa were excluded from the analysis ($r = -0.48$; $p < 0.05$).

Relationships between yield, N input, and N balance (the latter calculated as N input from fertilizer, manure, and fixation minus N removal) are of interest because N is typically the most limiting factor in rice cropping systems and also an important source of environmental pollution[30,31]. In general, a large positive N balance is a strong indicator of inefficient fertilizer use and potential reactive N losses into the environment, while a negative N balance suggests high risk of soil N mining that degrades soil quality[32]. For example, data from cereal systems show that

potential N losses increase substantially when N balance exceeds 75 kg N per ha[32–34]. Our analysis shows a positive linear relationship between the degree of yield gap closure and N input ($r = 0.75$; $p < 0.01$) and, to a lesser degree, with N balance ($r = 0.39$; $p < 0.05$) (Fig. 3A, B). Cropping systems with smallest yield gap tend to have N inputs and N balance $> 150$ and 50 kg N ha$^{-1}$, respectively, with a yield-scaled N balance ranging between zero and 20 kg N Mg$^{-1}$ grain (Fig. 3). Within the group of cropping systems with small yield gaps, some have a relatively small N balance (50–75 kg N ha$^{-1}$) and yield-scaled N balance ($<10$ kg N Mg$^{-1}$ grain) as in California and Australia (Fig. 3B, C). In contrast, other cropping systems with small yield gaps, such as southern USA and southern and central China, exhibit a

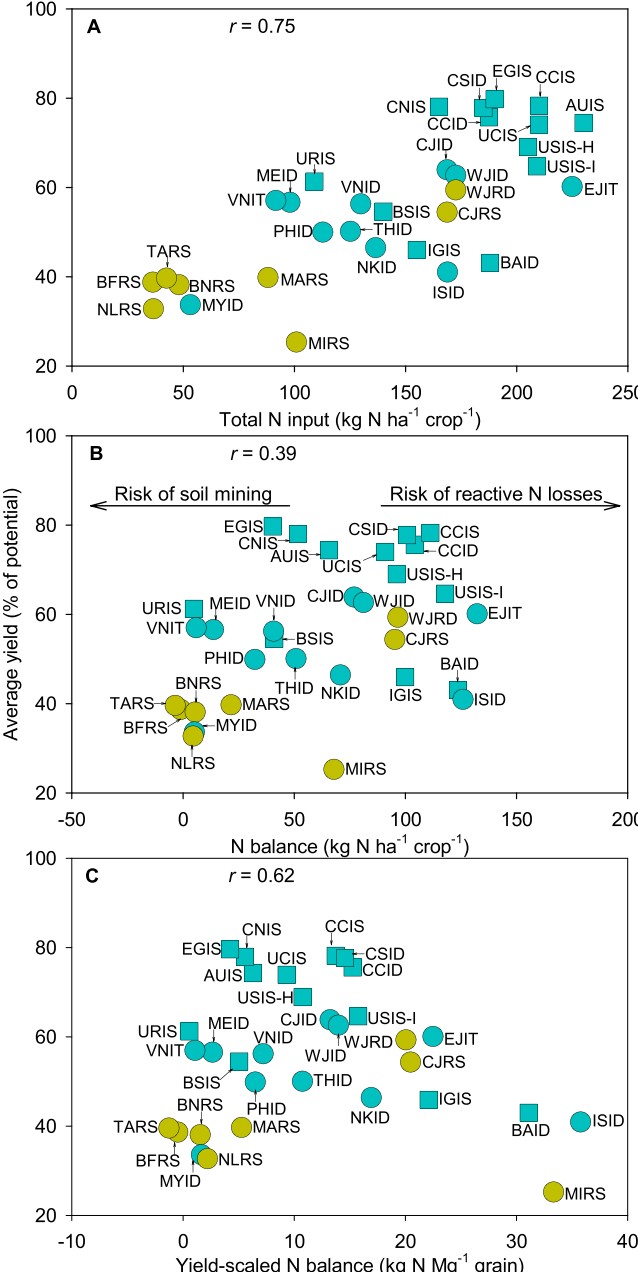

**Fig. 3 Yield, nitrogen (N) input and N balance across 32 rice cropping systems.** Panels show average rice yield (expressed as percentage of yield potential) versus **A** total N input (from fertilizer, manure, and fixation), **B** N balance calculated as N input minus N removal, and **C** yield-scaled N balance. Symbol type and color are used to distinguish tropical versus non-tropical regions (circles and squares, respectively) and irrigated versus rainfed systems (blue and yellow, respectively). Pearson's correlation coefficient (r) is shown only when the association between variables is significant (two-tailed Student's t-test; p < 0.01; n = 32 cropping systems). Cropping system codes are shown in the caption to Fig. 1. Data are provided in Source Data.

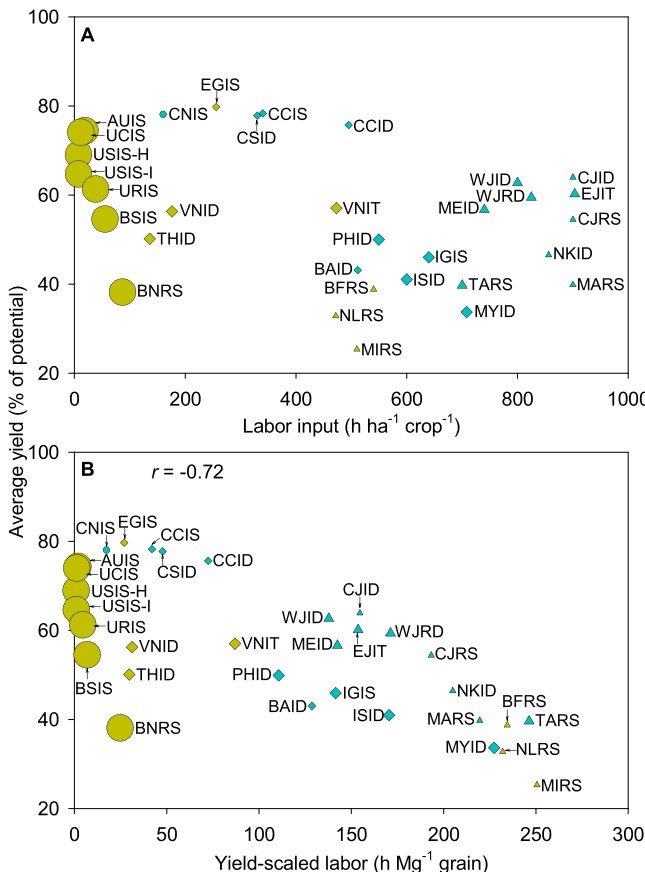

**Fig. 4 Yield and labor input across 32 rice cropping systems.** Panels show average rice yield (expressed as percentage of yield potential) versus **A** labor input per hectare and **B** yield-scaled labor. Symbols are used to distinguish systems with high (circle), intermediate (diamond), and low (triangle) level of mechanization. Symbol color indicates the predominant establishment method in each system: transplanting (blue) or direct seeding (yellow). Symbol size is proportional to the average field size in each system. Pearson's correlation coefficient (r) is shown only when the association between variables is significant (two-tailed Student's t-test; p < 0.01; n = 32 cropping systems). Cropping system codes are shown in Fig. 1. Data are provided in Source Data.

cropping systems in Sub-Saharan Africa exhibiting negative N balance, suggesting soil N mining over time (Fig. 3 and Supplementary Fig. 7). These systems would clearly benefit from larger N inputs or other methods to improve soil N supply. On the other hand, there is a group of systems with N input >150 N ha$^{-1}$ and large yield gaps, resulting in a large positive N balance on both area and yield-scaled basis, which is the case of several cropping systems in South and South-East Asia. In these systems, it seems feasible to reduce N inputs while maintaining yields or, perhaps more interestingly from a crop production perspective, to increase current yield with the same N input, in both cases leading to lower environmental impact and greater profit. This global analysis also shows that, while a small yield-scaled metric is desirable in the case of GWP, water, or pesticides, it is preferable that the (yield-scaled) N balance is maintained within an acceptable range (i.e., not excessively high or excessively low) to avoid both soil N mining and reactive N losses (Fig. 3C).

**Labor in rice cropping systems.** Labor use varies >100 times (ranging from 7 to 900 h ha$^{-1}$ crop$^{-1}$) across rice cropping systems, with the degree of mechanization explaining differences among

relatively large N balance (>100 kg N ha$^{-1}$) and yield-scaled N balance (>15 kg N Mg$^{-1}$ grain), suggesting room for reducing N input and N balance without yield penalty.

The relationship between average yield and yield-scaled N balance follows a curvilinear pattern (r = 0.62; p < 0.05), with larger yield gaps at both ends of the range of yield-scaled N balance (Fig. 3C). On the one hand, there are a number of

countries (Fig. 4A and Supplementary Table 5). Although it is difficult to separate cause-effect, the analysis suggests that large field size, high mechanization level, and direct seeding are intrinsically linked. For example, labor input is <40 h ha$^{-1}$ crop$^{-1}$ in highly mechanized systems in the USA, Australia, and Uruguay, where field size is >40 ha and rice is direct seeded (Fig. 4A and Supplementary Table 5). In contrast, labor input is >400 h ha$^{-1}$ crop$^{-1}$ (and up to 900 h ha$^{-1}$ crop$^{-1}$) in less mechanized systems such as those in Sub-Saharan Africa and Asia, where field size is <3 ha and most of the rice is transplanted.

One can still find large differences in yield-scaled labor (i.e., number of hours per unit yield) for a given labor input and there is a negative association between degree of yield gap closure and yield-scaled labor ($r = -0.72$; $p < 0.01$), which is consistent for both less mechanized and highly mechanized systems (Fig. 4B). For example, in the case of less mechanized systems, yield-scaled labor is smaller in South-East Asia and China (average: 110 h Mg$^{-1}$) compared to Sub-Saharan Africa (>200 h Mg$^{-1}$). Similar variation is observed across highly mechanized systems, with low yield-scaled labor in the USA and Australia compared to South America (1 versus 12 h Mg$^{-1}$). To summarize, our study shows no trade-offs between yield gap closure and labor requirements while yield-scaled labor decreased with smaller yield gaps in both labor-intensive and highly mechanized cropping systems. This finding suggests that a simultaneous improvement in yield and labor productivity is possible, which is relevant in the context of increasing labor wages and shrinking rural population in developing countries[35,36].

**Overall system performance**. We compute an overall performance index for the 32 rice cropping systems in our study (Fig. 5). Our analysis shows that the overall system performance is better in non-tropical versus tropical regions, probably due to inherent differences in soil and climate endowments leading to different resource-use efficiency and input requirements (e.g., higher nutrient and pesticide requirement per unit of yield in tropical environments)[37]. Still, one can identify systems that outperform other systems within each environment, as it is the case of California, Australia, and northern China (non-tropical regions), and Vietnam and Thailand (tropical regions). The analysis shown in Fig. 5 is also useful to identify, for a given country, where largest opportunities exist (yield gap, resource-use efficiency, or labor) to improve the overall performance of the cropping system. For example, pesticide use and N balance per unit of production is disproportionally higher in a number of cropping systems in South-East Asia and South Asia. Finally, disaggregation of the analysis at the level of each crop cycle can be useful to identify opportunities for improvement within each cropping system. For example, comparison of tropical cropping systems separately for the dry and wet seasons shows that Vietnam performs worse in the wet than in the dry season due to larger yield gap and higher yield-scaled number of pesticide applications (Supplementary Fig. 8).

## Discussion
Our global assessment of rice production systems evaluates use of arable land, water, energy, nutrients, pest control, and labor across a wide range of climates, soils, and water supply. Knowing the comparative advantage that a country has in terms of producing high and stable yields with high efficiency in use of required resources provides essential strategic insight to government agencies, international organizations, and charitable foundations (e.g., CGIAR, USAID, World Bank, UNEP, FAO, B&M Gates Foundation) for prioritizing investments on agricultural R&D at national to global scales. Our study also shows that an

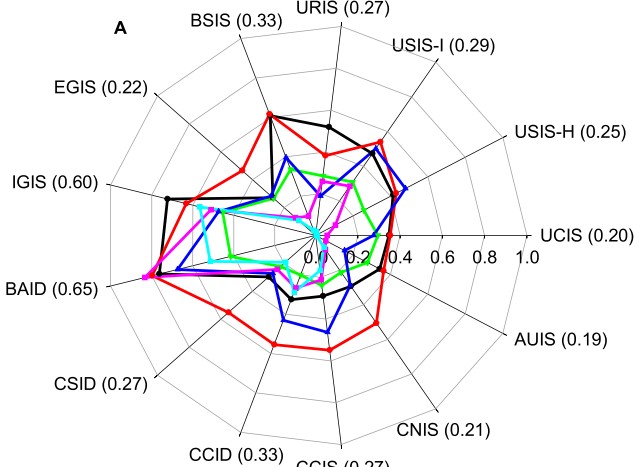

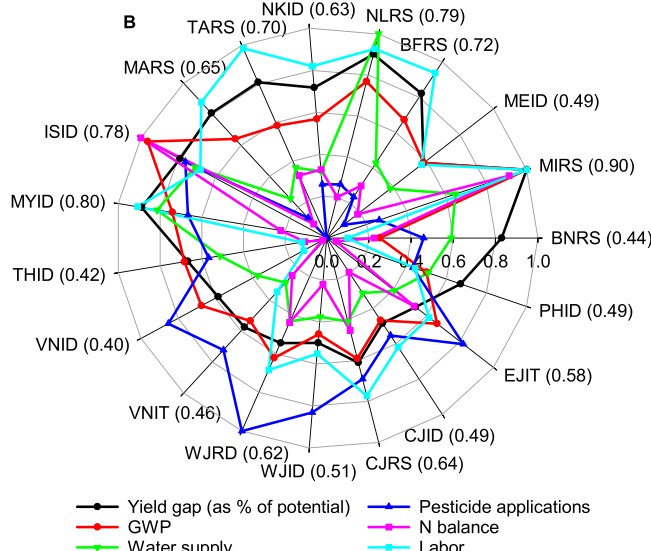

**Fig. 5 Comparison of yield gap (as percentage of yield potential) and yield-scaled metrics including global warming potential (GWP), water supply, number of pesticide applications, nitrogen (N) balance, and labor across 32 rice cropping systems.** Panels show radar charts for **A** non-tropical and **B** tropical regions. For each metric, data were normalized relative to the maximum value across all cropping systems, except for the yield-scaled N balance, which was expressed as an absolute deviation from 8 kg N Mg$^{-1}$ grain. Parenthetic values are the performance index of each system, with lower (higher) values indicating better (worse) overall performance. Cropping system codes are shown in caption to Fig. 1. See "Methods" section for explanation about the calculation of the overall performance index. Data are provided in Source Data.

explicit focus on areas with large yield gaps and/or large environmental impacts, complemented with economic and risk considerations, could help increase the return on investments in agricultural R&D programs. For example, increasing average yield to a level equivalent to 75% of the yield potential in 19 cropping systems where current yields are intermediate or low (<60% of yield potential) would increase global annual rice production by 146 Mt (+32% of current level) (Table 1), which would be sufficient to meet projected global rice demand by 2030[5]. Similarly, reducing the current large N balance (>100 kg N ha$^{-1}$) observed in eight cropping systems, so that N balance does not exceed 75 kg N ha$^{-1}$, could reduce the annual N excess by 2 Mt N, which is equivalent to a 95% reduction in the overall N excess across the 32 cropping systems. We also investigate, for the same set of 19

**Table 1 Potential production and environmental impact of closing yield gap to 75% of yield potential in 19 cropping systems with relatively large yield gaps (i.e., average yield < 60% of yield potential) with (i) simultaneous reduction of nitrogen (N) balance to 75 kg N ha⁻¹ in eight cropping systems with large N balance (>100 kg N ha⁻¹) and (ii) without changes in current yield-scaled N balance.**

| Scenario | Rice production (Mt)[a] | Excess N (Mt)[a] | |
|---|---|---|---|
| | | N balance reduction | Unchanged yield-scaled N balance |
| Baseline | 456 | 2.1 | 2.1 |
| Potential | 602 | 0.1 | 5.0 |
| Difference[b] | +146 (+32%) | −2 (−95%) | +2.9 (+140%) |

Excess N was calculated as the amount of N balance exceeding 75 kg N ha⁻¹. See "Methods" section for a description of the scenario assessment.
[a]Values are totals across the 32 rice cropping systems included in our analysis. For the potential scenario, we assumed no changes in current rice harvested area, cropping intensity, and proportion of irrigated area.
[b]Absolute and percentage difference between the potential scenario and current baseline.

cropping systems, what would be the environmental impact of closing the existing exploitable yield gap without changes in the current yield-scaled N balance. For this scenario, we project that annual reactive N excess would increase by 140% over the current level, totaling 5 Mt N (Table 1). Hence, our study highlights the importance of increasing both yield and resource-use efficiency, wherever room exists to do so, in order to produce sufficient rice while reducing the negative environmental impact and mitigating climate change. Accomplishing this dual goal will require not only agronomic interventions but also proper institutions and policies.

Current average yield has already reached 75% of the yield potential in a number of cropping systems, including Egypt, California, Australia, and China (Fig. 1). The level of yield gap closure in these systems suggests limited room for substantial yield increases, which is consistent with evidence of yield plateaus[6,7,38]. Efforts to increase current yield potential via improved cultivars, in concert with fine-tuning of current crop management, would likely provide small, but important, incremental improvements in the yield ceiling of those systems or, at least, to achieve modest yield gains by further closure of the existing yield gap[16,38,39]. Despite the positive relationship between yields and GWP, pesticides, N inputs, and labor, rice systems with small yield gaps tend to have lower yield-scaled resource use in comparison to other systems (Figs. 2–5). Hence, our study confirms that achieving high yields while minimizing negative environmental impact per unit of grain are not conflicting goals[23,40,41]. There are still cases of cropping systems with small yield gap but comparably poor resource-use efficiency, such as those in central China, where there is room to reduce the environmental impact while maintaining the same (high) yield level, which is consistent with empirical evidence from the literature[23,40,42,43].

At the other end of the spectrum, there are cropping systems where opportunities exist to narrow down the current (large) yield gap via higher use of fertilizers and/or pesticides[44,45], as it is the case of rice cropping systems in Sub-Saharan Africa and some countries in South-East Asia such as the Philippines and Myanmar (Figs. 2 and 3). Measures to promote higher agricultural inputs application in these countries must be accompanied with robust extension services and proper crop and soil management to fully capture the positive effect of improved plant health and nutrition and to minimize negative environmental impact[42,46].

Likewise, closing the exploitable yield gap requires breeding programs that release rice cultivars with improved tolerance to evolving pests and diseases and abiotic stresses such as submergence, salinity, and toxicities[47–49]. The yield gap is also large for upland rice in northern Brazil, which is a "transitional system" that starts as a low-input system after land conversion, shifting into a high-input soybean system a few years later[50]. We note that upland rice area has decreased sharply during past decades and its contribution to global rice production is small and less stable than in flooded lowland rice systems[9,51].

In between cropping systems with large and small yield gaps, there are a number of cropping systems that, for a given input level, exhibit consistently lower yield, suggesting room to produce more with current or less inputs. This is the case of many rice cropping systems in South and South-East Asia (e.g., Bangladesh, India, and Indonesia), where it may be possible to increase yields further and do so while reducing environmental impact (Figs. 2, 3, and 5). These cases require fine-tuning current crop, soil, and water management practices (and related policy) and a number of previous studies have shown how knowledge-intensive approaches can help increase yields and/or resource-use efficiencies, and, ultimately, improve farmers' profits[42,52,53]. Reducing production risks is also important to foster intensification[16]. This is the case for rainfed lowland rice systems, where farming is risk-prone because crops are more likely to be affected by drought, floods, pests, diseases, and weed outbreaks, as well as soil constraints[54]. Because of higher risk, most farmers in rainfed lowlands are reluctant to apply inputs (e.g., fertilizer) in similar amounts to irrigated lowlands[9]. As a result, rainfed lowland rice consistently exhibits larger yield gaps compared with irrigated rice at any level of water supply, and larger water use per unit of production (Fig. 2). Overall, with implementation of improved measures to mitigate risk (e.g., access to water pumps to apply partial irrigation during periods of water shortage, crop insurances), rainfed lowlands offer substantial room for increasing rice production because total rainfed lowland rice represents one-third of global rice harvested area[9].

To be effective at increasing the global rice output while reducing associated negative environmental impact, it is also important to consider the socio-economic context. The strong association between the degree of yield gap closure and national gross domestic product (GDP) per capita ($r = 0.68$; $p < 0.01$) suggests that farmers in systems with largest opportunities to increase yield (from a biophysical perspective) are at a disadvantage in terms of access to microfinance, inputs, mechanization, markets, and extension education services as it is the case in Sub-Saharan Africa (Supplementary Fig. 9). Similarly, options to increase yield and/or reduce environmental impact should consider potential trade-offs[25,55]. For instance, water-saving techniques look promising but could increase production risk and may be difficult to implement considering the level of sophistication in water and crop management that is needed[56]. Hence, while our study shows that there are opportunities to improve yield and/or resource-use efficiency in most rice cropping systems around the world, the means to achieve the desired level of sustainable intensification have to be tailored for each environment based upon the biophysical and socio-economic background.

## Methods
**Data sources.** Eighteen rice-producing countries were selected for our analysis (Supplementary Table 1). Those countries account for 88 and 86% of global rice production and harvested rice area[2], respectively (FAOSTAT, 2015–2017). We followed two steps to select the dominant cropping systems in each country. Within each country, our study focused on the main rice-producing area(s) (Supplementary Tables 2 and 3). For example, in the case of Brazil, we selected the southern and northern regions, which together account for nearly all rice

production in this country. In the case of Vietnam, we selected the Mekong Delta region, which accounts for nearly 60% of national rice production[57]. While we tried to cover all major rice cropping systems in each country, this was not possible in the case of rainfed lowland rice cropping systems in northeastern Thailand and eastern India because of lack of reliable estimates of yield potential and access to farmer yield and management data. Once the main rice-producing region(s) in each country was (were) identified, we then determined the dominant rice cropping system(s) for each of them (Supplementary Table 3). We note that "cropping system" refers to a unique combination of a number of rice crops planted on the same piece of land within a 12-month period (and their temporal arrangement), water regime (rainfed or irrigated), and ecosystem (upland or lowland) (Supplementary Fig. 1 and Supplementary Table 2). In our study, rice cropping systems are single-, double-, or triple-season rice; none of the cropping systems are ratoon rice. Following the previous examples, two cropping systems were selected for Brazil (rainfed upland single rice and lowland irrigated single rice in the northern and southern regions, respectively) and two systems were selected for the Mekong Delta region in Vietnam. These systems account for nearly all rice harvested areas in these regions. We distinguished between rice-based cropping systems sowing hybrid versus inbred cultivars in the southern USA. Across the 18 countries, this study included a total of 32 rice cropping systems, which, in turn, covered 51% of the global rice harvested area (Supplementary Tables 1 and 3). Note that the area coverage reported here corresponds to that accounted for 32 cropping systems (and not by the countries where the cropping systems were located). These systems portrayed a wide range of biophysical and socio-economic backgrounds (Supplementary Figs. 1 and 2 and Supplementary Tables 1 and 2), leading to average rice yields ranging from 2–10.4 Mg ha$^{-1}$ (Supplementary Fig. 3). For data analysis purposes, rice cropping systems were classified into tropical and non-tropical[9,58,59] and also based upon water regime and crop season.

Agronomic information was collected via structure questionnaires completed by agricultural specialists in each country or region (Supplementary Table 6). The collected data included field size, tillage method, crop establishment method, degree of mechanization for each field operation, seeding rate, crop establishment, and harvest dates, nutrient fertilizer rates, manure type, and rate, pesticides (number of applications, products, and rates), irrigation amount (in irrigated systems), energy source for irrigation pumping, labor input, and straw management (Supplementary Tables 4 and 5). Average values for each cropping system reported by country experts were retrieved from survey data available from previous projects (Supplementary Table 7). Rice grain yield was reported at a standard moisture content of 140 g H$_2$O kg$^{-1}$ grain, separately for each crop cycle, using data from, at least, three recent cropping seasons in each cropping system. In the case of irrigated rice cropping system in Nigeria and Mali, data were only available for one crop cycle in double-season rice. In this case, we assumed management and actual yield to be identical in the two cycles.

In all cases, and wherever possible, data were cross-validated with other independent datasets (e.g., FAOSTAT, World Bank, IFA, and published journal papers), which gives confidence about the representativeness and accuracy of the survey data. For example, we estimated area-weighted national yield according to actual yield provided for each cropping system and annual rice harvested area in each system for each of the 18 countries. Comparison of these yields against those reported by FAOSTAT[2] showed a strong association and agreement between data sources (Supplementary Fig. 10). We also cross-validated actual yield, N fertilizer, labor, and irrigation from our database with those reported by previous studies (published after the year 2000) based on on-farm data collected in ten selected countries. Due to the lack of on-farm data on irrigation, we used published data collected from experiments that follow typical farmer irrigation practices. In the case of irrigation, our cross-validation differentiated between crop seasons (wet versus dry) in the case of irrigated double-season rice cropping systems. In all cases, average yield, N fertilizer, labor, and irrigation from our database fell within (or very close) the range of values reported in previously published studies for those same cropping systems (Supplementary Table 8). Measured daily weather data, including daily solar radiation, minimum and maximum temperatures, and precipitation, were derived from representative weather stations in each region (Supplementary Fig. 2 and Supplementary Table 9). Data on per-capita gross domestic product (GDP) during 2015–2017 were retrieved for each country to explore relationships between yield gap and economic development[60] (Supplementary Fig. 9 and Supplementary Table 1).

**Estimation of yield gaps.** The yield gap is defined as the difference between yield potential and average farmer yield. Estimates of yield potential for irrigated rice or water-limited yield potential for rainfed rice were adopted from Global Yield Gap Atlas (GYGA)[61] (Supplementary Table 7). Yield potential simulation in GYGA was performed using crop growth and development model ORYZA2000 or ORYZA (v3) (except for APSIM in the case of India) and based on actual data on crop management, soil data, measured daily weather data, and representative rice varieties planted in each region (see details for yield potential simulation in Supplementary Information Text Section 1). Data on yield potential were not available for Australia (AUIS) in GYGA; hence, we used estimates of yield potential from Lacy et al.[62]. Yield potential (or water-limited yield potential for rainfed rice) and average yields were computed separately for each rice crop in each rice cropping system (Supplementary Fig. 3). The coefficient of variation (CV) of yield potential

(or water-limited yield potential) was estimated for each cropping system (Supplementary Fig. 4). In this study, average rice yield was expressed as percentage of the yield potential (or water-limited yield potential for rainfed rice) for each cropping system (Fig. 1 and Supplementary Fig. 5). In those cropping systems where more than one rice crop is grown within a 12-month period, we estimated potential and average yield on both per-crop and annual basis by averaging and summing up the estimates for each rice crop, respectively. In the case of per-crop averages, for those cropping systems in which the harvested rice area changed between crop cycles, we weighted the values for each cycles based on the associated harvested rice area. However, for simplicity, the main text reports only the values on a per-crop basis; annual estimates are provided in the Supplementary Information. Normalizing average yield by the yield potential at each site provides a direct comparison of yield gap closure across systems with diverse biophysical backgrounds (e.g., variation in solar radiation, temperature, and water supply). Without this normalization, one might make biased assessment in relation to the available room for improving yield. For example, an actual yield of 8 Mg ha$^{-1}$ is equivalent to 80% of yield potential in the cropping system of central China, whereas a yield of 8 Mg ha$^{-1}$ achieved by irrigated rice farmers in Brazil only represents 55% of yield potential (Supplementary Fig. 3).

**Quantifying resource-use efficiency.** We assessed the performance of rice production by calculating the following metrics: global warming potential (GWP), fossil-fuel energy inputs, water supply (irrigation plus in-season precipitation), number of pesticide applications, nitrogen (N) balance, and labor input, each expressed on an area and yield-scaled basis (Figs. 2, 3 and 4 and Supplementary Figs. 6, 7 and 11). We estimated metrics on both per-crop and annual basis and report the values on a per-crop basis in the main text while the annual estimates are provided in the Supplementary Information. In the case of GWP, it includes CO$_2$, CH$_4$, and N$_2$O emissions (expressed as CO$_2$-eq) from (i) production, packaging, and transportation of agricultural inputs (seed, fertilizer, pesticides, machinery, etc.), (ii) fossil-fuel energy directly used for farm operations (including irrigation pumping), and (iii) CH$_4$ and N$_2$O emission during rice cultivation[63]. Emissions from agricultural inputs were calculated on application rates and associated GHG emissions factors (see details in Supplementary Information Text Section 2, Supplementary Table 10). In the case of fossil fuel used for field operations, it was calculated based on the number and type of farm operations and associated fuel requirements (Supplementary Table 11). Total N$_2$O emissions were calculated as the sum of direct and indirect N$_2$O emissions. A previous meta-analysis including rice showed that direct soil N$_2$O emissions can be estimated from the magnitude of N-surplus, which was calculated as applied N inputs minus accumulated N in aboveground biomass at physiological maturity[21]. Therefore, direct soil N$_2$O emissions for a given rice crop cycle were estimated following van Groenigen et al. N-balance approach[21]. Indirect N$_2$O emissions were estimated based on the Intergovernmental Panel on Climate Change (IPCC) methodology[64], assuming indirect N$_2$O emissions represent 20% of direct N$_2$O emissions. The CH$_4$ emissions from rice paddy field were calculated following IPCC[65]. Following this approach, CH$_4$ emissions are estimated considering the duration of the rice cultivation period, water regime during the cultivation period and during the pre-season before the cultivation period, and type and amount of organic amendment applied (e.g., straw, manure, compost) based on a baseline emission factor. We assumed no net change in soil carbon stocks as soil organic matter is typically at steady state in lowland rice[66]. We did not attempt to estimate changes on soil C in the upland rice system in Brazil. All emissions were converted to CO$_2$-eq, with GWP for CH$_4$ set at 25 relatives to CO$_2$ and 298 for N$_2$O on a per mass basis over a 100-year time horizon[67]. For each rice crop cycle in each of the 32 rice systems, GWP was calculated as the sum of CO$_2$, CH$_4$, and N$_2$O emissions expressed as CO$_2$-eq. (Details on N$_2$O and CH$_4$ emissions estimates and GWP calculations are provided in Supplementary Information Text Section 2).

Calculation of energy inputs was similar to that of GWP and was based on the reported rates of agricultural inputs and field operations and associated embodied energy (see details for energy input estimates in Supplementary Information Text Section 2, Supplementary Table 12). Human labor was also included in the calculation of energy inputs. There was a strong positive relationship between energy input and GWP on both per-crop ($r = 0.81$; $p < 0.01$) and annual basis ($r = 0.92$; $p < 0.01$), so we only showed results on GWP in the main text. Results on energy input and net energy yield (the difference between energy output and input) on a per-crop or annual basis can be found in Supplementary Fig. 11.

The N balance was calculated as the difference between N input from synthetic N fertilizer, manure, and biological N fixation minus N removal with the harvested grain (and straw if it was burned or removed from the field) following Dobermann and Witt[68] (see details for N balance estimates in Supplementary Information Text Section 3). The N input and N removal were estimated for each rice crop cycle. The N input via manure was calculated based on the amount and source of manure and N concentration. An average input of N from biological N fixation of 30 kg N ha$^{-1}$ crop$^{-1}$ was assumed for lowland rice systems[69], while biological N fixation in upland rice was assumed to represent 10% of that in lowland rice[70]. Grain N removal was calculated based on average grain yield and rice grain N concentration. The N removal with straw was estimated assuming a typical percentage of straw remaining in the field and percentage of N lost from the crop residue in different straw managements (Supplementary Table 13). In our N

balance calculation, we assumed N losses via lixiviation and denitrification to be similar to the amount of N inputs via irrigation water and atmospheric deposition[52]. A threshold of N balance of 75 kg N ha$^{-1}$ was used in this study to estimate N excess (and potentially large reactive N losses), as potential N losses increase substantially when N balance exceeds 75 kg N ha$^{-1}$ as reported by previous studies[32–34].

We estimated the amount of active ingredient and environmental impact quotient (EIQ) of pesticides including insecticide, herbicide, and fungicide applied per hectare per crop following Kovach et al. environmental risk assessment methodology[71] (see details for toxicity estimation in Supplementary Information Text Section 4). The two metrics showed a significant and positive relationship on a per-crop basis (Supplementary Information Text Section 4, $r = 0.96$; $p < 0.01$), and EIQ was also significantly and positively correlated with the number of pesticide applications ($r = 0.87$; $p < 0.01$). Given the uncertainty in EIQ estimates associated with sketchy reporting of products and application rate of pesticides, and considerable variation in the reliability of such data among countries or regions, the number of pesticide applications is used to evaluate environmental risk among cropping systems.

**Labor requirement**. Labor requirement is a key driver explaining changes in rice area, systems, and profit[63,72]. Our labor data included labor involved in land preparation, seed preparation, crop establishment, water irrigation, fertilization, pesticide application, weeding, harvesting, threshing, and drying (Supplementary Table 4). Given the intrinsic relationships among labor input, mechanization level, establishment method (direct seeding versus transplanting), and field size[72], we characterized each rice cropping system in terms of these parameters and expressed labor input on both area and yield-scaled basis (see details for labor input and degree of mechanization in Supplementary Information Text Section 5, Fig. 4 and Supplementary Fig. 6; Supplementary Table 4).

**Estimation of overall performance index**. We computed a semi-quantitative index to quantify the performance of each cropping system in relation to six metrics, including the yield gap (as percentage of yield potential) and yield-scaled metrics including GWP, water supply, number of pesticide applications, N balance, and labor input (Fig. 5 and Supplementary Fig. 8). For each metric, the score was calculated by normalizing the data relative to the maximum value among all 32 cropping systems. An exception was the yield-scaled N balance, which was expressed as an absolute deviation from 8 kg N Mg$^{-1}$ grain. This value corresponds to the average yield-scaled N balance estimated for Australia and California, which we assumed here to be a reasonable target to achieve the dual goal of minimizing the N balance and closing the yield gap, while avoid soil N mining (Fig. 3). Finally, we estimated an overall performance index for each rice cropping system by averaging the individual scores associated with the six metrics. Four out of the six metrics are related with yield-scaled metrics (GWP, water supply, number of pesticide applications, and N balance), one with yield gaps, and another one with labor. To avoid biases, we weighted each individual score so that yield gap, resource-use efficiency, and labor will have a similar impact on the computation of the overall performance index. Lower (higher) overall index indicates better (worse) overall performance (Fig. 5). Following previous assessments of sustainability in cropping systems[25,55,73–75], radar charts were used to show the performance of each cropping system in this study in relation to yield-scaled metrics, yield gaps, and labor. Separate analyses were performed based upon climate background (non-tropical and tropical) and also by crop season (wet and dry) in the case of tropical environments (Fig. 5 and Supplementary Fig. 8). Finally, Pearson's correlation coefficients were calculated to investigate associations between resource-use efficiency and yield gaps (Supplementary Table 14). Statistix 8 and SigmaPlot 12.5 were used for statistical analysis.

**Scenario analysis**. To illustrate the potential of our assessment to serve as basis to prioritize agricultural R&D, we explored a scenario in which there is an explicit effort to (i) increase average yield from current level to 75% of yield potential in cropping systems with relatively large yield gaps (defined here as average yield < 60% of yield potential), and (ii) reduce current N balance to 75 kg N ha$^{-1}$ crop$^{-1}$ in cropping systems that currently exhibit a large N balance (defined here as N balance > 100 kg N ha$^{-1}$ crop$^{-1}$). Selection of this N balance target (i.e., 75 kg N ha$^{-1}$ crop$^{-1}$) was based on data from the literature showing that large N losses occur when N balance exceeds that value[32–34]. Following these criteria, we selected a total of 19 cropping systems with large gaps (mostly in Sub-Saharan Africa and Asia) and eight cropping systems with large N balance (mostly in China and South Asia). Following previous studies, the excess N was calculated as the amount of N balance exceeding 75 kg N per ha[76]. We also explored a second scenario in which, for the same set of 19 systems, average yield increases from current level to 75% of yield potential without changes in current yield-scaled N balance (Table 1). Because the goal of these two scenarios was to understand how to produce more while reducing the environmental impact on existing global rice area, we calculated the potential extra rice production and changes in excess N across the 32 cropping systems considering current rice harvested area, cropping intensity, and proportion of irrigated area (Table 1).

**Uncertainty and limitations**. We acknowledge the uncertainty related with estimation of yield potential and collection of actual yield and management data. In all cases, we used estimates of yield potential derived using well-calibrated models and best available sources of weather, soil, and management data. To the extent that it was possible, we cross-validated estimates of yield potential with measured yield data collected from well-managed crops that grew without nutrient limitation and without yield reductions due to biotic stresses. In the case of survey data, there is always uncertainty in relation to the representativeness of the regions and years included for the analysis. The analyses presented herein focused on the most intensively cropped area of each cropping system using data from at least three cropping seasons for each area. We note, however, that we could not include drought-prone rainfed lowland rice cropping systems in northeastern Thailand and eastern India in our analysis due to lack of robust estimates of yield potential and data on inputs and management. Hence, our findings for rainfed lowland rice apply to those systems in Sub-Saharan Africa and Indonesia. Future work should include these cropping systems. While variation still exists within each of cropping system in terms of crop sequence configuration, management, and inputs, we also note that some level of spatial aggregation was needed for the purpose of the cross-system comparison presented in this paper and also to be effective at orienting agricultural R&D at the national and regional levels. Similarly, to make a fair comparison of the cropping systems and for interpretation of the results, we have to categorize cropping systems according to water regime (irrigated versus rainfed), climate background (tropical and non-tropical), cropping intensity (single, double, and triple), crop season (dry and wet), and, in the case of labor, establishment method (transplanting versus direct seeding). Our categorization can be further improved, for example, by adding other biophysical and socio-economic factors that can help interpret the results and design interventions. Likewise, estimation of GWP required a number of assumptions in relation to GHG emissions; in those cases, we relied on the most recent literature to derive appropriate emission factors. Overall, our assessment should be considered as an initial step, which could be further refined as more spatially granular data on yield, inputs, and agronomic management, and methods to estimate GHG emissions, become available in the future. However, we do not expect these sources of uncertainty and limitations to affect the conclusions from this study. Detailed description of data sources and estimations of yield gaps, resource-use efficiency, and labor inputs can be found in Supplementary Information.

**Reporting summary**. Further information on research design is available in the Nature Research Reporting Summary linked to this article.

## Data availability
Data on yield potential from Global Yield Gap Atlas are available at www.yieldgap.org. Data on per-capita gross domestic product from the World Bank are available at https://databank.worldbank.org. Data on rice distribution from SPAM are available at www.mapspam.info. Data on rice area and production from FAOSTAT are available at www.fao.org/faostat. We note that questionnaire participants gave full consent to share the data used in our study. Source data files are provided with this paper. Source data are provided with this paper.

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

## Acknowledgements

We would like to thank Dr. Russell Ford (former Head of Agronomic R&D at Sunrice) for providing data for rice in Australia and Dr. P.A.J. van Oort for performing the simulations of yield potential for African countries. We would also like to thank scientists and extension personnel for their help to collect the survey data from the 32 cropping systems included in this study. This work was supported by the Major International (Regional) Joint Research Project of NSFC (32061143038 to S.P.), the Earmarked Fund for the China Agriculture Research System (CARS-01-20 to S.P.), the Program of Introducing Talents of Discipline to Universities in China (the 111 Project no. B14032 to S.P.), the China Scholarship Council (201706760015 to S.Y.), and the China Postdoctoral Science Foundation (2020M682439 to S.Y.). We also acknowledge GRISP, RICE CRP, and the Swiss Agency for Development and Cooperation (Grant 681 no. 7F-08412.02 to A.M.S.) for their financial support to conduct the MISTIG, MISTIR, and CORIGAP surveys, respectively.

## Author contributions

S.Y., K.G.C., S.P., and P.G. conceived and designed the study. B.A.L., L.T.W., A.M.S., V.P., B.M., K.S., N.A., V.E.A., L.Y.K., A.J.Z., A.B.H., G.C., N.S., P.S.B., T.L., and S.P. provided the data analyzed in this study. S.Y. and P.G. compiled the data, performed the data analysis, and wrote the paper. All authors contributed to editing the paper.

## Competing interests

The authors declare no competing interests.
