## [Peer Review File · Nature Communications]

Reviewer's Comments

Paper: A roadmap towards sustainable intensification for a larger global rice bowl

Manuscript ID: NCOMMS-21-13531

Rice is one of the most important crops in the world, feeding almost half of the whole world population. Producing enough rice to the increasing population is a priority in many countries in the world. Yield gap has been the central concept in crop science and sustainable intensification for a long time. This paper assessed both yield gap and resource-use efficiency (including water, pesticides, nitrogen, labor, and energy) across major rice cropping systems of the world. The authors found that most cropping systems have room for increasing yield, resource-use efficiency, or both. They concluded that current total rice production of these systems can be increased by 36%, and excess nitrogen almost eliminated, by focusing on a relatively small number of cropping systems with large yield gaps and/or poor resource-use efficiencies. This indeed could be valuable insight for policy makers to prioritize the investment and effort in expanding and improving rice production.

The paper obviously deal with a critical challenge of agriculture and the findings are also worthwhile. However, I do have a few major concerns.

1. My first concern is the datasets used. This is a global study, covering all the major rice systems in the world (the authors say it accounts for 88% of global rice production). The yield gap is well documented in Global Yield Gap Atlas project. However, the other datasets, e.g. greenhouse gas emission, energy inputs, pesticides, labor, are complex and the authors have not explained the exact data sources. Indeed the SI file has detailed sections on how these parameters are estimated but left lot of details unanswered. We even could not know how the 32 rice cropping systems are classified and defined. Are they really representative of the global rice production? These are the consensus systems recognized by rice experts? For example, each of the 32 rice cropping systems covers a large, heterogeneous region. How do you account for the spatial heterogeneity within each system? North America only has 3 systems: UCIS, USIS-I, USIS-H. What are the boundaries for these 3 systems? How are they classified? The actual rice production is much more complex and the variations across regions are huge. How are your greenhouse gas emission, energy inputs etc

calculated in such diverse, large regions? They are averaged across country data or weighted average from state-level (in US) or department-level (in Mexico) data? Or they are estimated from representative station data points? Either way of aggregation/averaging, there are lot of questions on the credibility, accuracy, and representativeness of the final estimated datasets.

2. My second concern is the concept of closing yield gap. Closing yield gap is much more complex than just finding the gaps, or assuming achieving 75% of yield potential as done in this paper (Line 145 in this paper). Closing yield gap is not about if any region or subregions (the 32 rice systems) have the theoretic possibility (which we all know they do). Rather it is about how realistically it could be achieved. These need cost-benefit analysis, policy changes, and adoptions of technologies such as advanced seed, fertilizer, pesticide, irrigation etc. Farmers are in fact very smart entrepreneurs. It is not that they don't want to have higher yield (e.g. closing yield gap). The current yield level, which is far lower than the potential yield in most regions, is the optimum for them, given the current policy (e.g. price, market, regulations), constraints (e.g. credits, access to knowledge and extension), and benefit/cost ratios. Recent research has shown that this "economic yield gap" is much smaller than the agronomic yield gap, and it (the economic yield gap) even doesn't exist in many regions. In this sense, the brave claim of increasing rice production by 36% by achieving 75% of yield potential is far from the reality.
3. My third concern is the presentation and easy reading of the paper. While the paper is generally well-written, the paper is hard to follow and to read. Particularly the large number of abbreviations for the rice systems: CNIS, EJIT, USIS etc are the core of the paper and they are all over the figures and tables. It is hard for even an expert to really distinguish them and comprehend these tables/figures. It may be OK for a small field journal. And yet for a widely circulated journal like Nature Communications, the extra requirement of clearly presentation of technical contents and appeal to non-experts is justified.

Reviewer #2 (Remarks to the Author):

A roadmap towards sustainable intensification for a larger global rice bowl

Yuan and colleagues conduct a global analysis of rice production system in order to investigate the association between crop yield (assessed as yield gap) and selected sustainability indicators for greenhouse gas emissions, water use, pesticides, labor, energy, and N balance. The scale of the analysis is coarse, with either national level or broad sub-national regions defined as 'cropping systems' for the purposes of analysis. The yield gap assessment component of the study aggregates existing country reports from the GYGA online portal. This study is timely, with the discourse around what constitutes sustainable agriculture very heated. The only way forward is to aggregate evidence around key performance indicators to assess / manage potential tradeoffs, to set achievable development goals, and to establish R&D priorities that are responsive to geographic differences.

Beyond summarizing global-scale information on the rice yield gap from the GYGA, the authors of the study endeavor to address three main questions:

- 1) How are differences in rice productivity associated with environmental and other sustainability outcomes?
- 2) Can performance benchmarks be set for different cropping systems?
- 3) Where and how should R&D investment be targeted to support sustainable intensification and maximize ROI?

In general, the paper is well-written and analytical choices justified and appropriate. There are, however, some areas for attention that would likely strengthen the study and its presentation. Here I highlight some major points; additional technical comments and questions to be considered are included in the annotated version of the PDF.

#1 Cropping system characterization data: beyond the YG estimates, this study hinges on the quality of the expert opinions mobilize to characterize all other facets of the studied rice systems including fertilizer use, the field water balance, residue management, labor investment, pesticide use etc. The authors acknowledge the uncertainties of this approach, but don't offer any qualitative validation of the results to ensure they are at least directionally correct. The process of developing and verifying these data should be more fully described. When checked against secondary data, where the results reliable? The community needs much more of these types of data collected at scale to both document 'means' and their spatial distribution. Moreover, through direct experience, I suspect that there is a tremendous amount of management and site variation within almost all of these cropping systems. As such, the authors should ensure that the strengths and limitations of the approach for priority setting at a coarse spatial scale are adequately considered.

#2 Data analysis and graphics: Figures 2, 3, and 4 use scatter plots to visualize the relationships between different sustainability indicators (e.g. water input) and the yield gap. What is the justification for using yield gap as the dependent variable instead of yield itself? Since there is a very significant amount of Yp variation in the crop systems studied, why should there be a strong relationship between input / efficiency terms and the yield gap per se?

Also, there are likely clearer ways to visualize these analyses. For example, wouldn't a more parsimonious single figure substitute for Fig 2A and 2B by simply graphing yield-scaled GWP as the dependent variable against yield? This would establish an efficiency frontier at different yield levels while also graphically demonstrating the generalized association between yield intensification and key sustainability indicators.

#3 Benchmarking: The analysis then attempts to establish what's possible (i.e. aspiration goals) for rice systems in Figure 5 (radar charts) and by computing a overall performance index for each cropping system. Rice growing environments are incredibly diverse, and I am bit concerned that all are being judged on the same scale. The most straightforward critique of this approach would be with respect to water. In monsoonal climates, there are substantial water fluxes (i.e. deep percolation, overland flow) that cannot be controlled by farmers. Should these fluxes be considered inefficiencies? Setting benchmarks that are more relevant to production ecology differences would seem to be a prudent step. An initial typology for more realistic benchmark standards could segregate dry and wet season metrics as a first step.

#4 Scenario analysis: The current scenario analysis demonstrates that by closing yield gaps in selected countries and addressing excess N application in others, progress towards development and food security goals can be achieved. In many senses this is not new information. Perhaps a more nuanced scenario analysis would take advantage of the sustainability analysis and ask about the implications of increasing yields without transforming sustainability indicators. That is, what are the costs of maintaining the status quo in rice production systems with room for growth?

#5 Establishing priorities: The authors state that "This study provides essential strategic insight for prioritizing national and global agricultural R&D investments to ensure adequate rice supply while minimizing negative environmental impact in coming decades". The broad scope of the study's findings are broadly known with respect to prevailing yield gaps and also with respect to cropping systems with overuse of inputs. To meet the 'roadmaps' expectation of the title, I encourage the authors to consider more deeply what their results suggest for both the geographies and types of development interventions that are likely to contribute to sustainable intensification.

Reviewer #3 (Remarks to the Author):

Judgment criteria for Nature Communications are:

1. What are the noteworthy results?
2. Will the work be of significance to the field and related fields? How does it compare to the established literature? If the work is not original, please provide relevant references.
3. Does the work support the conclusions and claims, or is additional evidence needed?
4. Are there any flaws in the data analysis, interpretation and conclusions? Do these prohibit publication or require revision?
5. Is the methodology sound? Does the work meet the expected standards in your field?
6. Is there enough detail provided in the methods for the work to be reproduced?

1. The authors provided useful information for policy makers, governments and international organizations on the current situation on global food security (the productivity gap between the ideal and the reality) in line with SDG1 and SDG2. Sustainable intensification is getting more important for the continued support to projects on agriculture, so additional discussion on resource-use efficiency would provide valuable information on SDG6 and SDG15. The authors could emphasize the climate action (SDG13; climate change mitigation) a bit more, with minimal revision.

2. The work reported here combined the outputs of rice yield simulations using the existing freeware (ORYZA) to get the ideal yield and on-farm survey (but mostly published/reported elsewhere) from questionnaires to get the actual yield. Each step has no research originality as it is. The outputs of this study would be more suitable for the global statistical yearbook such as FAOSTAT rather than for a scientific article. However, there has been no attempt to compile both to come up with the rice productivity gap at the global scale. Data standardization (data cleaning) across countries with close coordination among scientists from different institutions would not be easy, which deserves publication.

3. The title does not match what the authors showed and discussed; a roadmap for sustainable rice intensification sounds broad and ambiguous. The authors can use more direct phrase such as

food security, less hunger or closing the gap highlighting rice productivity increase in a sustainable manner.

4. The major concern in the analysis is about the ideal productivity under rainfed lowland conditions. I appreciate the authors' great effort to get the standard (stable) yields by averaging the data over three seasons. However, unlike rainfed wheat and maize in uplands, hydrological conditions (i.e. water availability) cannot be simply estimated/predicted for rainfed lowland rice by soil and rainfall as shown by a number of papers (e.g. toposequential effect, a large inter-field variation in water availability). In addition, there was no data from drought-prone rainfed lowland rice in Asia; we know rainfed lowland rice in Java is mostly favorable as seen in Fig. 2C (little difference in water supply between rainfed and irrigated fields). Our biggest challenge in closing the yield gap lies in rainfed agriculture or difficult rice environments where, to date, there's no best way to properly estimate the ideal yield (water-limitation mode in ORYZA cannot precisely capture the yield potential in rainfed lowland fields). In this regard, "(the current situation in) 88% of global rice production" was covered by this study is too exaggerated and misleading (e.g. rainfed lowlands in Northeast region is the major rice bowl in Thailand, which was not included in the discussion for Thailand).

Nevertheless, these criticisms would not deny the value of the very first attempt for global rice yield gap by the authors. It should be specified that the authors have not yet covered the large portion of rainfed lowland areas prone to abiotic stresses such as Indo-China and Eastern India in the preliminary version, and that the future work should focus on such ecosystems. The authors' discussion on rainfed lowland ecosystem (Line 415 and onwards) was not supported/evidenced by the data of this study.

5. Rice yield declared by farmers (through questionnaires) is often different from that measured by destructive samplings (plant science-based measurements). But the ideal yield is based on the plant science (crop simulation models). We know quick and accurate yield estimation is a big challenge, please elaborate the authors' statement "yield data were cross-validated with other independent datasets". Likewise, please elaborate the measurement of irrigation amount (the authors assumed the water flow from inlets (how?) and asked farmers of the hours of irrigation?) and labour input (how could the authors estimate work hours for each process?).

6. The authors can prudently think about the use of present tense or past for each sentence to improve readability. Continued effort on rice breeding for high yield with traits of resource-use efficiency should be more emphasized (Line 318 and onwards). Also, development of new tolerant cultivars for (a)biotic stresses itself will greatly narrow the yield gap, the importance for breeding effort should be emphasized.

Figure 5 just compiled the raw data from previous figures. Is the radar chart the only style in the synthetic analysis? Please consider PCA or cluster to categorize them into a few major components, as it seems the authors were very conscious about the categorization in the Discussion part (e.g. Line 394).

Editorial suggestions:

- (1) The current manuscript is voluminous, please make it more concise.
- (2) Rainfed upland rice and rainfed lowland rice (not upland rainfed rice or lowland rainfed rice) throughout the manuscript.
- (3) "Larger environmental impact (Line 178)", where majority of existing lands have been already reclaimed for rice cultivation, ecologists for biodiversity conservation may not like such exaggeration because each field has a limited (i.e. finite) capacity to buffer the effects of agrochemical inputs. In this case, land area basis cannot be changed.
- (4) "where crops are likely to be fully irrigated (Line 223)" may not be enough to tell the story. The careful choice of fields (soil type; clay) for rice cultivation to minimize the percolation is also involved, often regulated by local governments.
- (5) "simultaneous improvement in crop and labour productivity (Line 315) needs a good agricultural mechanization program. Please discuss the significance of mechanization program and/or microfinance for smallholder farmers in LDC.

Responses to Reviewers' Comments (and our responses in red)

Comments from Editor

As you will see from the reports copied below, the reviewers raise important concerns. Without substantial revisions, we will be unlikely to send the paper back to review. In particular, concerns about the definition of cropping systems and yield gap reduction. I would like to remind you that the Method section is not taken into account for the word limit and so it can be extended as necessary to clarify the methodology used. For the rest of the manuscript, please prioritise providing the information needed in response to reviewers suggestions rather than omitting information to make the manuscript concise.

Answer: We thank the three reviewers for their comments on our paper and their overall support for publication after revision. We have addressed all their comments in the revised MS. As suggested by the editor, we have extended the methods section to provide further details on the methodology. Our point-by-point responses to reviewers' comments are shown below and changes in the revised MS are marked in red. We have also submitted the PDF file provided by Reviewer #2 with his/her detailed comments, and our answers to them, as a related MS file.

Comments from Reviewer #1

Rice is one of the most important crops in the world, feeding almost half of the whole world population. Producing enough rice to the increasing population is a priority in many countries in the world. Yield gap has been the central concept in crop science and sustainable intensification for a long time. This paper assessed both yield gap and resource-use efficiency (including water, pesticides, nitrogen, labor, and energy) across major rice cropping systems of the world. The authors found that most cropping systems have room for increasing yield, resource-use efficiency, or both. They concluded that current total rice production of these systems can be increased by 36%, and excess nitrogen almost eliminated, by focusing on a relatively small number of cropping systems with large yield gaps and/or poor resource-use efficiencies. This indeed could be valuable insight for policy makers to prioritize the investment and effort in expanding and improving rice production.

Answer: Thanks for highlighting the value of our study.

The paper obviously deal with a critical challenge of agriculture and the findings are also worthwhile. However, I do have a few major concerns. My first concern is the datasets used. This is a global study, covering all the major rice systems in the world (the authors say it accounts for 88% of global rice production). The yield gap is well documented in Global Yield Gap Atlas project. However, the other datasets, e.g. greenhouse gas emission, energy inputs, pesticides, labor, are complex and the authors have not explained the exact data sources. Indeed the SI file has detailed sections on how these parameters are estimated but left lot of details unanswered. We even could not know how the 32 rice cropping systems are classified and defined. Are they really representative of the global rice production? These are the consensus systems recognized by rice experts? For example, each of the 32 rice cropping systems covers a large, heterogeneous region. How do you account for the spatial heterogeneity within each system? North America only has 3 systems: UCIS, USIS-I, USIS-H. What are the boundaries for these 3 systems? How are they classified? The actual rice production is much more complex and the variations across regions are huge. How are your

greenhouse gas emission, energy inputs etc calculated in such diverse, large regions? They are averaged across country data or weighted average from state-level (in US) or department-level (in Mexico) data? Or they are estimated from representative station data points? Either way of aggregation/averaging, there are lot of questions on the credibility, accuracy, and representativeness of the final estimated datasets.

Answer: Thanks for these comments. We fully agree with the reviewer and recognize the importance of credibility and accuracy of the dataset used in our study to compute GHG emissions, input-use efficiency, and other parameters.

First, we note that GHG emissions, energy, and labor values were derived from primary data on inputs and management practices, following well-established methods explained in the original MS (Sections 2-5 of Supplementary Information). So, we did not further elaborate on explaining these methods in the revised MS.

Second, in response to reviewer's comments on cropping system selection and data sources, and also in connection to editor's comment on cropping system definition, we have elaborated further in the Methods section of the revised MS about how dominant rice cropping systems were identified (L 451-475 of revised MS). We also added the rice area accounted by each of the selected regions as well as the area accounted for by each of the selected cropping system in relation to the total rice area in each region to show that our selected areas and systems are highly representative of the rice systems in each country (Table S3 of revised SI).

Third, we agree with Reviewer #1 that there is a spatial heterogeneity in rice systems within each region. We note that our original MS already accounts for some of this variation. For example, our analysis included five different rice systems in Indonesia, three in the USA, and four in China. While we agree that variation exists within each of these systems in terms of crop sequence configuration, management, and inputs, we also note that some level of spatial aggregation is needed for the purpose of the cross-system comparison presented in this paper and also to be effective at orienting agricultural R&D at national and regional level. Having said that, our assessment should be considered as an initial step, which could be further refined as more spatially granular data on yield, inputs, and agronomic management become available in the future. Based on Reviewer #1 comments, we added text elaborating on the limitations of our study (L 687-691 & L 699-702 of revised MS).

Finally, in relation to data collection and representativeness, we added text on how data were collected (L 451-475 of revised MS). Briefly, we identified the main rice producing region(s) in each country and, subsequently, we identified the most dominant cropping system(s) for each of the selected regions. We did our best effort to collect data from areas that were representative from each of these systems as determined by rice experts in each country. In an effort to provide confidence about the representativeness of the selected cropping systems and data sources, we cross-validated the area-weighted national average yield based on actual yield provided for each cropping system against national official statistics in each country and the comparison shows that yield was fairly similar between two data sources, which gives us confidence on the data used for our assessment. Furthermore, our data on yield, N rate, and labor were cross-validated with those reported in previously published studies based on on-farm survey (L 496-510 of revised MS, Fig. S10 & Table S7 of revised SI). Due to unavailability of on-farm data on irrigation, we compared our data against those reported in experiments that followed farmers' practice (Table S7 of revised SI). We also elaborated on

the uncertainty associated with data collection in the discussion section (L 687-691 & L 699-702 of revised MS). We thank Reviewer #1 for this comment as it pushed us to further validate our database and give evidence that our data can be considered representative of the cropping systems that we are evaluating in our paper.

My second concern is the concept of closing yield gap. Closing yield gap is much more complex than just finding the gaps, or assuming achieving 75% of yield potential as done in this paper (Line 145 in this paper). Closing yield gap is not about if any region or subregions (the 32 rice systems) have the theoretic possibility (which we all know they do). Rather it is about how realistically it could be achieved. These need cost-benefit analysis, policy changes, and adoptions of technologies such as advanced seed, fertilizer, pesticide, irrigation etc. Farmers are in fact very smart entrepreneurs. It is not that they don't want to have higher yield (e.g. closing yield gap). The current yield level, which is far lower than the potential yield in most regions, is the optimum for them, given the current policy (e.g. price, market, regulations), constraints (e.g. credits, access to knowledge and extension), and benefit/cost ratios. Recent research has shown that this "economic yield gap" is much smaller than the agronomic yield gap, and it (the economic yield gap) even doesn't exist in many regions. In this sense, the brave claim of increasing rice production by 36% by achieving 75% of yield potential is far from the reality.

Answer: Thanks for this comment. Understanding the current yield gap and resource-use efficiency is one step for informing agricultural R&D investments but it is not sufficient. It must be complemented with socio-economic data. For example, it may be difficult to narrow down yield gaps in areas where gaps are larger but farmers lack reasonable access to inputs, markets and extension services, as it is the case of farmers in Sub-Saharan Africa (L 435-438 of revised MS). Having said that, it does not preclude an analysis of additional rice production from closing the exploitable yield gap in these regions for a scenario in which socio-economic constraints are alleviated. Hence, we do not see a conflict between the message delivered by our MS and the comment made here by Reviewer #1. We added text in relation to this issue to our section on uncertainty and limitations (L 366-369, L 695-697 & L 699-702 of revised MS).

My third concern is the presentation and easy reading of the paper. While the paper is generally well-written, the paper is hard to follow and to read. Particularly the large number of abbreviations for the rice systems: CNIS, EJIT, USIS etc are the core of the paper and they are all over the figures and tables. It is hard for even an expert to really distinguish them and comprehend these tables/figures. It may be OK for a small field journal. And yet for a widely circulated journal like Nature Communications, the extra requirement of clearly presentation of technical contents and appeal to non-experts is justified.

Answer: Thanks for this comment. We realized that some text of the results section fits better in the Supplementary Material, which, in turn, helps improve the flow of the main text. Based on reviewer's comments, we moved the entire second paragraph of the results section to the Supplementary Material (L 94-108 of revised SI). We also streamlined the text in other paragraphs of the results sections. After careful consideration, we decided to keep acronyms in the figures as in the original MS. We simply could not find a better way to show results from a cross-system comparison without some system of acronyms that would allow readers to identify each system in each figure, which is crucial for interpretation of the results.

Comments from Reviewer #2

Yuan and colleagues conduct a global analysis of rice production system in order to investigate the association between crop yield (assessed as yield gap) and selected sustainability indicators for greenhouse gas emissions, water use, pesticides, labor, energy, and N balance. The scale of the analysis is coarse, with either national level or broad sub-national regions defined as ‘cropping systems’ for the purposes of analysis. The yield gap assessment component of the study aggregates existing country reports from the GYGA online portal. This study is timely, with the discourse around what constitutes sustainable agriculture very heated. The only way forward to is to aggregate evidence around key performance indicators to assess / manage potential tradeoffs, to set achievable development goals, and to establish R&D priorities that are responsive to geographic differences.

Beyond summarizing global-scale information on the rice yield gap from the GYGA, the authors of the study endeavor to address three main questions:

- 1) How are differences in rice productivity associated with environmental and other sustainability outcomes?
- 2) Can performance benchmarks be set for different cropping systems?
- 3) Where and how should R&D investment be targeted to support sustainable intensification and maximize ROI?

In general, the paper is well-written and analytical choices justified and appropriate. There are, however, some areas for attention that would likely strengthen the study and its presentation. Here I highlight some major points; additional technical comments and questions to be considered are included in the annotated version of the PDF.

Answer: We thank the reviewer for the overall support to our study and his/her helpful comments. We note that Reviewer #2 has also submitted a PDF file with specific comments and edits; these comments were addressed in an appended PDF file, submitted as related MS file. We thank Reviewer #2 for the very detailed review of the MS.

#1 Cropping system characterization data: beyond the YG estimates, this study hinges on the quality of the expert opinions mobilize to characterize all other facets of the studied rice systems including fertilizer use, the field water balance, residue management, labor investment, pesticide use etc. The authors acknowledge the uncertainties of this approach, but don't offer any qualitative validation of the results to ensure they are at least directionally correct. The process of developing and verifying these data should be more fully described. When checked against secondary data, where the results reliable? The community needs much more of these types of data collected at scale to both document ‘means’ and their spatial distribution. Moreover, through direct experience, I suspect that there is a tremendous amount of management and site variation within almost all of these cropping systems. As such, the authors should ensure that the strengths and limitations of the approach for priority setting at a coarse spatial scale are adequately considered.

Answer: See our previous response to Reviewer #1 about this same issue. Briefly, we have elaborated further in the Methods section of the revised MS about how dominant rice cropping systems were identified (L 451-475 of revised MS). We also added the rice area accounted by each of the selected regions as well as the area accounted for by each of the selected cropping system in relation to the total rice area in each region to show that our selected areas and systems are highly representative of the rice systems in each country (Table S3 of revised SI). Similarly, we included a cross-validation for some key variables in

our database, including yield, fertilizer, labor, and irrigation (L 496-510 of revised MS, Fig. S10 & Table S7 of revised SI).

#2 Data analysis and graphics: Figures 2, 3, and 4 use scatter plots to visualize the relationships between different sustainability indicators (e.g. water input) and the yield gap. What is the justification for using yield gap as the dependent variable instead of yield itself? Since there is a very significant amount of Y_p variation in the crop systems studied, why should there be a strong relationship between input / efficiency terms and the yield gap per se? Also, there are likely clearer ways to visualize these analyses. For example, wouldn't a more parsimonious single figure substitute for Fig 2A and 2B by simply graphing yield-scaled GWP as the dependent variable against yield? This would establish an efficiency frontier at different yield levels while also graphically demonstrating the generalized association between yield intensification and key sustainability indicators.

Answer: Thanks for these comments. Our figures plot the average farm yield, expressed as percentage of the yield potential. Normalizing average yield by the yield potential at each site is needed for a fair comparison of the level of yield gap closure across systems with diverse biophysical background (e.g., variation in solar radiation, temperature, and water supply). Without this normalization, one might make biased assessment in relation to the available room for improving yield. For example, an actual yield of 8 Mg ha^{-1} is equivalent to 80% of yield potential in the cropping system of central China, whereas a yield of 8 Mg ha^{-1} achieved by irrigated rice farmers in Brazil only represents 55% of yield potential. Following reviewer's comments, we elaborate further on the justification for using yield as % of yield potential rather than actual yield (L 138-139 & L 538-545 of revised MS). We also believe that it is important to show variables on both an output- and area-basis. For example, someone may conclude that the potential environmental impact is larger with higher input and yield levels per unit land; however, the figures showing yield-scaled metrics that, indeed, GHG, water, pesticide and N balance per unit of production decrease with higher level of intensification. Hence, we prefer to keep Figure 2 as it is now, also considering that other reviewers did not make suggestions to modify that figure.

#3 Benchmarking: The analysis then attempts to establish what's possible (i.e. aspiration goals) for rice systems in Figure 5 (radar charts) and by computing a overall performance index for each cropping system. Rice growing environments are incredibly diverse, and I am bit concerned that all are being judged on the same scale. The most straightforward critique of this approach would be with respect to water. In monsoonal climates, there are substantial water fluxes (i.e. deep percolation, overland flow) that cannot be controlled by farmers. Should these fluxes be considered inefficiencies? Setting benchmarks that are more relevant to production ecology differences would seem to be a prudent step. An initial typology for more realistic benchmark standards could segregate dry and wet season metrics as a first step.

Answer: We thank the reviewer for pointing out this issue. We acknowledge that comparing rice systems is challenging given the diverse background. We did some categorization in the original MS (e.g., irrigated and rainfed crops, tropical and non-tropical environments, seeding method in the case of labor) but we agree that it may not be sufficient to fully interpret some of the findings. In response to Reviewer #2 comment, we did a further categorization based on cropping season (dry and wet) (L 327-332 & L 649-650 of revised MS and Fig. S8 of revised SI). We also included text indicating that a more granular categorization could be desirable for a better interpretation of the results in our 'limitations and uncertainty' section

(L 691-697 & L 699-702 of revised MS).

#4 Scenario analysis: The current scenario analysis demonstrates that by closing yield gaps in selected countries and addressing excess N application in others, progress towards development and food security goals can be achieved. In many senses this is not new information. Perhaps a more nuanced scenario analysis would take advantage of the sustainability analysis and ask about the implications of increasing yields without transforming sustainability indicators. That is, what are the costs of maintaining the status quo in rice production systems with room for growth?

Answer: Following the reviewer's suggestion, we explored an alternative scenario in which average yield is increased from current level to 75% of yield potential without changes in yield-scaled N balance (L 362-365 & L 665-668 & Table 1 of revised MS). We thank the Reviewer #2 for his excellent suggestion, which helped us extend our scenario analysis.

#5 Establishing priorities: The authors state that "This study provides essential strategic insight for prioritizing national and global agricultural R&D investments to ensure adequate rice supply while minimizing negative environmental impact in coming decades". The broad scope of the study's findings are broadly known with respect to prevailing yield gaps and also with respect to cropping systems with overuse of inputs. To meet the 'roadmaps' expectation of the title, I encourage the authors to consider more deeply what their results suggest for both the geographies and types of development interventions that are likely to contribute to sustainable intensification.

Answer: We thank the reviewer for this comment. Because both Reviewers #2 and #3 (see below) indicated some disconnection between the title of the article and the delivered results, we modified the title (see L 1 of revised MS).

Comments from Reviewer #3

Judgment criteria for Nature Communications are:

1. What are the noteworthy results?
2. Will the work be of significance to the field and related fields? How does it compare to the established literature? If the work is not original, please provide relevant references.
3. Does the work support the conclusions and claims, or is additional evidence needed?
4. Are there any flaws in the data analysis, interpretation and conclusions? Do these prohibit publication or require revision?
5. Is the methodology sound? Does the work meet the expected standards in your field?
6. Is there enough detail provided in the methods for the work to be reproduced?

1. The authors provided useful information for policy makers, governments and international organizations on the current situation on global food security (the productivity gap between the ideal and the reality) in line with SDG1 and SDG2. Sustainable intensification is getting more important for the continued support to projects on agriculture, so additional discussion on resource-use efficiency would provide valuable information on SDG6 and SDG15. The authors could emphasize the climate action (SDG13; climate change mitigation) a bit more, with minimal revision.

Answer: Done (L 366-368 of revised MS).

2. The work reported here combined the outputs of rice yield simulations using the existing freeware (ORYZA) to get the ideal yield and on-farm survey (but mostly published/reported elsewhere) from questionnaires to get the actual yield. Each step has no research originality as it is. The outputs of this study would be more suitable for the global statistical yearbook such as FAOSTAT rather than for a scientific article. However, there has been no attempt to compile both to come up with the rice productivity gap at the global scale. Data standardization (data cleaning) across countries with close coordination among scientists from different institutions would not be easy, which deserves publication.

Answer: We thank the Reviewer #3 for the support for publication.

3. The title does not match what the authors showed and discussed; a roadmap for sustainable rice intensification sounds broad and ambiguous. The authors can use more direct phrase such as food security, less hunger or closing the gap highlighting rice productivity increase in a sustainable manner.

Answer: See our previous response to Reviewer #2. Briefly, we changed the title to better reflect the contents of the article (L 1 of revised MS).

4. The major concern in the analysis is about the ideal productivity under rainfed lowland conditions. I appreciate the authors' great effort to get the standard (stable) yields by averaging the data over three seasons. However, unlike rainfed wheat and maize in uplands, hydrological conditions (i.e. water availability) cannot be simply estimated/predicted for rainfed lowland rice by soil and rainfall as shown by a number of papers (e.g. toposequential effect, a large inter-field variation in water availability). In addition, there was no data from drought-prone rainfed lowland rice in Asia; we know rainfed lowland rice in Java is mostly favorable as seen in Fig. 2C (little difference in water supply between rainfed and irrigated fields). Our biggest challenge in closing the yield gap lies in rainfed agriculture or difficult rice environments where, to date, there's no best way to properly estimate the ideal yield (water-limitation mode in ORYZA cannot precisely capture the yield potential in rainfed lowland fields). In this regard, "(the current situation in) 88% of global rice production" was covered by this study is too exaggerated and misleading (e.g. rainfed lowlands in Northeast region is the major rice bowl in Thailand, which was not included in the discussion for Thailand). Nevertheless, these criticisms would not deny the value of the very first attempt for global rice yield gap by the authors. It should be specified that the authors have not yet covered the large portion of rainfed lowland areas prone to abiotic stresses such as Indo-China and Eastern India in the preliminary version, and that the future work should focus on such ecosystems. The authors' discussion on rainfed lowland ecosystem (Line 415 and onwards) was not supported/evidenced by the data of this study.

Answer: Following reviewer's suggestion, we added text to indicate that we did not cover some important rainfed lowland rice systems such as northeast Thailand and eastern India and that future work should include these ecosystems (L 683-687 of revised MS). We adjusted the rice area coverage accordingly (L 111-112 & L 472-475 of revised MS and Table S3 of revised SI). We note that despite rainfed lowland rice in India and Thailand was not included, our assessment includes rainfed lowland rice production in Indonesia and Sub-Saharan Africa. Hence, the discussion of rainfed lowland rice is still valid for these systems. In response to this comment, we also added text to the 'uncertainty and limitations' section to make sure that our conclusion is valid only for the rainfed lowland rice systems included in

our assessment (L 685-686 of revised MS).

5. Rice yield declared by farmers (through questionnaires) is often different from that measured by destructive samplings (plant science-based measurements). But the ideal yield is based on the plant science (crop simulation models). We know quick and accurate yield estimation is a big challenge, please elaborate the authors' statement "yield data were cross-validated with other independent datasets". Likewise, please elaborate the measurement of irrigation amount (the authors assumed the water flow from inlets (how?) and asked farmers of the hours of irrigation?) and labour input (how could the authors estimate work hours for each process?).

Answer: See our previous response to Reviewer #1 on data sources. We added a cross-validation of yield and input variables including fertilizer, labor, and irrigation in Methods section (L 496-510 & L 621-623 of revised MS, Fig. S10 & Table S7 of revised SI).

6. The authors can prudently think about the use of present tense or past for each sentence to improve readability. Continued effort on rice breeding for high yield with traits of resource-use efficiency should be more emphasized (Line 318 and onwards). Also, development of new tolerant cultivars for (a)biotic stresses itself will greatly narrow the yield gap, the importance for breeding effort should be emphasized.

Answer: Following reviewer's comments, we chose the present tense and stick to it in the main text (except for the Methods section where we used past tense). We agree that the importance of breeding needs to be better emphasized and we have added text to the revised MS following the reviewer's suggestion (L 403-406 of revised MS).

Figure 5 just compiled the raw data from previous figures. Is the radar chart the only style in the synthetic analysis? Please consider PCA or cluster to categorize them into a few major components, as it seems the authors were very conscious about the categorization in the Discussion part (e.g. Line 394).

Answer: We thank the reviewer for this comment. We checked previously published works related with sustainability indicators and found that it seems a kind of standard to use radar chart in such analysis (e.g., Arnés et al., 2013; Arouna et al., 2021; Devkota et al., 2019; Greer et al., 2020; Pittelkow et al., 2016). Hence, we are inclined to leave the radar charts as they are now, but we added text to justify our choice and cited the aforementioned references (L 646-648 of revised MS).

Editorial suggestions:

(1) The current manuscript is voluminous, please make it more concise.

Answer: See our previous response to Reviewer #1 on this. Briefly, we have moved an entire paragraph from the results section to the text section of Supplementary Information (L 94-108 of revised SI) and streamlined the text in several parts of the revised MS.

(2) Rainfed upland rice and rainfed lowland rice (not upland rainfed rice or lowland rainfed rice) throughout the manuscript.

Answer: Following reviewer's suggestion, we changed the use of "upland rainfed rice" and "lowland rainfed rice" throughout the revised MS.

(3) “Larger environmental impact (Line 178)”, where majority of existing lands have been already reclaimed for rice cultivation, ecologists for biodiversity conservation may not like such exaggeration because each field has a limited (i.e. finite) capacity to buffer the effects of agrochemical inputs. In this case, land area basis cannot be changed.

Answer: We apologize if the original text was not clear. We meant that, in many cases, low-yield systems are located in areas where land resources exist for further rice production (e.g., Sub-Saharan African and South America) and, therefore, if pressure exists to increase rice production, it can lead to conversion of fragile natural ecosystems such as wetlands and forests. We rephrased the sentence for clarity (L 179-182 of revised MS).

(4) “where crops are likely to be fully irrigated (Line 223)” may not be enough to tell the story. The careful choice of fields (soil type; clay) for rice cultivation to minimize the percolation is also involved, often regulated by local governments.

Answer: We agree with the reviewer. We modified the text in the revised MS (L 212-213 of revised MS).

(5) “simultaneous improvement in crop and labour productivity (Line 315) needs a good agricultural mechanization program. Please discuss the significance of mechanization program and/or microfinance for smallholder farmers in LDC.

Answer: We added text following reviewer’s suggestion (L 437 of revised MS).

Reviewers' Comments:

Reviewer #1:

Remarks to the Author:

Congratulations on the substantial revisions of the paper. I'm happy with the revisions and your detailed responses to my comments. I recommended the acceptance of your paper.
Congratulations!

Reviewer #2:

Remarks to the Author:

My comments have been adequately addressed by the authors in their revised manuscript. In my view, it is ready for publication in Nature Communications and will make a valuable contribution to the literature.

We appreciate the strong support of the reviewers for publication. We have revised the paper accepting all the editorial suggestions. Our responses are shown below in red.

REVIEWERS' COMMENTS

Reviewer #1 (Remarks to the Author):

Congratulations on the substantial revisions of the paper. I'm happy with the revisions and your detailed responses to my comments. I recommended the acceptance of your paper. Congratulations!

Response: thanks

Reviewer #2 (Remarks to the Author):

My comments have been adequately addressed by the authors in their revised manuscript. In my view, it is ready for publication in Nature Communications and will make a valuable contribution to the literature.

Response: thanks